# Fabry Disease and Central Nervous System Involvement: From Big to Small, from Brain to Synapse

**DOI:** 10.3390/ijms24065246

**Published:** 2023-03-09

**Authors:** Elisenda Cortés-Saladelafont, Julián Fernández-Martín, Saida Ortolano

**Affiliations:** 1Unit of Inherited Metabolic Diseases and Neuropediatrics, Department of Pediatrics, Hospital Universitari Germans Trias i Pujol, 08916 Badalona, Spain; 2Department of Pediatrics, Facultat de Medicina, Universitat Autònoma de Barcelona, Campus Can Ruti, 08916 Badalona, Spain; 3Rare Diseases & Pediatric Medicine Research Group, Galicia Sur Health Research Institute (IIS Galicia Sur), SERGAS-UVIGO, 36312 Vigo, Spain; 4Internal Medicine Department, Complejo Hospitalario Universitario de Vigo, 36312 Vigo, Spain

**Keywords:** Fabry disease, synapse, neurotransmitter, neurotransmission, lysosome, metabolism

## Abstract

Fabry disease (FD) is an X-linked lysosomal storage disorder (LSD) secondary to mutations in the *GLA* gene that causes dysfunctional activity of lysosomal hydrolase α-galactosidase A and results in the accumulation of globotriaosylceramide (Gb3) and globotriaosylsphingosine (lyso-Gb3). The endothelial accumulation of these substrates results in injury to multiple organs, mainly the kidney, heart, brain and peripheral nervous system. The literature on FD and central nervous system involvement is scarce when focusing on alterations beyond cerebrovascular disease and is nearly absent in regard to synaptic dysfunction. In spite of that, reports have provided evidence for the CNS’ clinical implications in FD, including Parkinson’s disease, neuropsychiatric disorders and executive dysfunction. We aim to review these topics based on the current available scientific literature.

## 1. Introduction

Fabry disease (FD; MIM: 301500) is a lysosomal storage disorder (LSD) with an X-linked inheritance secondary to mutations in the *GLA* gene (NCBI: NC_000023.11; Xq22), which results in the absent or decreased activity of lysosomal hydrolase α-galactosidase A (AGA, α-GalA; BRENDA: EC3.2.1.22). The consequent accumulation of its primary substrate globotriaosylceramide (Gb3) and its derivatives (mainly globotriaosylsphingosine (lyso-Gb3)) results in injury to multiple organs and systemic dysfunction, with endothelial vascular involvement being the main pathological alteration in the disease [1]. Due to the X-linked mode of inheritance, men are generally more severely affected and disease manifestations occur earlier compared with women, for whom progression of the disease to organ failure generally occurs later in life, and in whom symptom severity tends to be milder and more variable than in males [2,3]. The same occurs for children and adolescences, since clinical manifestations appear earlier in boys than in girls [4]. Nevertheless, since the X chromosome is randomly inactivated in the cells of female patients, a severe phenotype can also be expressed in women [5]. In addition, a distinction is made between classical and non-classical disease phenotypes [2].

Although FD is among the most prevalent lysosomal disorders, with up to 2.5 cases per 100,000 males [6], little is known or has been reported regarding the involvement of the central nervous system (CNS) beyond cerebrovascular disease. This is particularly striking since there are studies available relating FD to certain neuropsychological profiles, Parkinson’s disease (PD), the accumulation of Lewy bodies, and movement or psychiatric disorders.

The aim of this article was to provide a state-of-the-art review of the available scientific publications relating FD to the CNS, with a special focus on its neuronal, synaptic and neurotransmitter implications. The conceptual line of the review aims to start with a wide focus based on the clinical manifestations reported in patients with FD (beyond those related to peripheral nervous system involvement and cerebrovascular disease), subsequently narrow the focus and describe the findings from magnetic resonance imaging, neuropathology and animal models, and conclude with what is known thus far about cellular metabolism and synaptic dysfunction in FD (see Figure 1 for a graphical summary of the text). For details on reference selection method (Appendix A) and search results (Appendix A), see Appendix A. 

## 2. The Framework of LSDs and Fabry Disease

Approximately two-thirds of patients suffering from LSDs are expected to exhibit some kind of neurological involvement, with a very different range of symptoms and a wide clinical spectrum [6,7]. All these characteristics and the neurological phenotypes have been more deeply explored and described for some other LSDs than FD. This is supported by the fact that the main clinical phenotype of FD differs widely from mucopolysaccharidosis III type C or Niemann–Pick type C disease, for example, in which neurodegeneration and dementia are the most prominent symptoms. In general, neurological manifestations of FD include small fiber neuropathy, associated with pain and reduced temperature sensation, and premature cerebrovascular events, attributed to complex vasculopathy secondary to progressive glycosphingolipid accumulation in vessels and to cardiac involvement [8]. Neurodegeneration and neuroinflammation manifesting with microgliosis and astrocytosis are described as the most common hallmarks of brain pathology in neurological LSD, and FD is not considered among them. With FD, the accumulation of Gb3/lyso-Gb3 could induce an inflammatory response and increase oxidative stress and apoptosis in different cell types with prominent endothelial damage (such as cardiomyocytes or podocytes) [1]. Moreover, substrate accumulation has increasingly been shown to involve cellular structures beyond the lysosome, and its downstream effects (fibrosis, inflammation, generation of reactive oxygen species, etc.) also seem to play key roles in pathogenesis [9].

## 3. Clinical Manifestations Related to CNS Involvement in FD

The classic neurological manifestations of FD are considered to be small fiber neuropathy associated with pain, along with reduced temperature sensation and heat intolerance, and premature cerebrovascular disease [10,11]. Both are attributed to progressive glycosphingolipid accumulation in vessels and to cardiac involvement, which leads to secondary complex vasculopathy. Ischemic stroke and transient ischemic attacks may occur, and white matter lesions are commonly seen on MRI, even in asymptomatic patients [12]. Autonomic nervous system disease may manifest with gastrointestinal symptoms similar to irritable bowel syndrome and hypohidrosis [13]. In the next sections, clinical manifestations involving the CNS and those beyond these classical manifestations are discussed and reviewed.

### 3.1. The Pain in Fabry Disease

The precise pathomechanism of neuropathic pain in FD is unknown, and since it might essentially be considered as part of the peripheral nervous system, it is beyond the scope of this review. Nevertheless, considering that pain in Fabry disease has a peripheral component and/or a possible central/medullar component (yet largely unexplored), the current review includes some lines on this topic [14].

The peripheral neuropathy in Fabry disease manifests as neuropathic pain and is one of the main hallmarks of the disease. It is characterized by reduced cold and warm sensation, gastrointestinal disturbances, burning pain with a globe and socking distribution, and sudden pain crises [11]. The pain is mainly localized in the hands and feet and encompasses fingers, palms and soles [15]. It can manifest as early as 2 or 3 years of age, is reported in both boys and girls, and is often associated with febrile illnesses, with reduced heat and exercise tolerance. It is a debilitating condition that negatively interferes in quality of life in these patients [16].

As a brief summary, the four main mechanisms linked to peripheral neuropathic pain in Fabry disease would be as follows: Glycolipid and Gb3 deposits in the perineurium, the endothelial cells (vasa nervorum), the dorsal root ganglion, and the Schwann cells, as well as dysfunctional ion channels in the nerves [15,17]. Focusing on the central mechanisms of neuropathic pain, it is important to distinguish (1) the central component, mainly due to direct injuries or sequela to the central nervous system (mostly identified in patients with multiple sclerosis, strokes, etc.), from (2) the central sensitization, which refers to a situation in which chronic nociceptive afferent input from a peripheral pain generator causes reversible changes in central nociceptive pathways [14]. The latter has been reported in Fabry disease; several studies report hypersensitivity to mechanical stimuli [15], and a central disinhibition pain mechanism due to a reduced A-delta fiber input (reduced protective sensitive afferent stimuli, not properly mediated by these A-delta fibers, leading to a constant C-fiber unmyelinated-input that mediates pain) [17]. When talking about this central mechanism, it has to be noted that basic central networks are involved including mechanisms of reward and antireward via the medial thalamic pathway.

### 3.2. Parkinson’s Disease (PD)

The prevalence of LSD mutations in PD patients strengthens the idea that lysosomal dysfunction is a key player in PD pathogenesis [9]. A sequencing analysis of 54 genes in PD patients that are causative of different LSDs were analyzed (including the *GLA* gene), and a total of 54% of the PD patients were shown to have at least one variant in one of those genes [18]. Moreover, studies have shown a direct relationship between decreased or loss-of-function activity of lysosomal glucocerebrosidase (accumulated in Gaucher disease) and effects on the processing and clearance of α-synuclein that facilitates its aggregation. In addition, impaired autophagy and lysosomal function were bi-directionally detected in many LSDs, including FD, and in PD, suggesting a possible link between these disorders [19,20,21]. AGA activity was also tested in PD patients and showed similar results, in that these patients presented lower activity of this enzyme than controls [22]. It is also of importance to mention that despite not being able to prove a neurodegenerative pattern in FD [8], motor abnormalities involving slower gait, reduced hand speed, and poorer fine manual dexterity have been identified and shown to be independent of cerebrovascular symptoms.

### 3.3. Neurodegeneration

A study with a large cohort of FD patients (110 patients: 60 heterozygous females, 50 hemizygous males) found that FD was associated with impaired motor function and various nonmotor symptoms, but that it did not lead to a pattern of extrapyramidal symptoms, significant cognitive problems, or other symptoms commonly preceding neurodegenerative diseases (PD or dementia with Lewy bodies) [8]. Nevertheless, the authors concluded that they could not rule out neurodegeneration in FD, since the study was focused only on detecting the clinical prodromes that normally precede other neurodegenerative disorders, for example, Gaucher disease. The authors stated that FD might lead to a focused brainstem pathology, resulting in a distinct clinical phenotype with mild motor impairment and nonmotor symptoms (i.e., depression, pain, daytime sleepiness, and hearing loss) but not associated with the cardinal clinical prodromes of neurodegenerative diseases. In contrast, in a report from one severely affected FD patient with a prominent hypokinetic phenotype, severe neuronal loss in the substantia nigra pars compacta and Lewy pathology was found [23]. Therefore, it might be concluded that larger cohorts of patients, with matching pathology findings and clinical manifestations, are also needed to analyze if a GLA deficit may contribute to the more severe phenotype in CNS when associated with other genetic or epigenetic factors.

### 3.4. Psychiatric Manifestations and Cognitive Functioning

Psychological and psychiatric manifestations, in particular, depression, have been reported to be common in FD. These patients also score significantly worse than general population samples and patients with other chronic diseases (including Gaucher disease, another LSD) on measures of depression, anxiety and health-related quality of life perception [24]. In a recent study from a Dutch cohort, patients scored significantly worse in terms of subjective cognitive status [25]. In the previously mentioned study from Löhle et al., reporting the prospective results from a large cohort of FD patients, they were able to reproduce what other studies had previously reported: A high prevalence of depression (up to 46%), pain and daytime sleepiness (up to 50–60%) in FD patients [26].

There is an interesting study reporting on four individuals from two generations from the same family, three of whom exhibited mainly neuropsychological symptoms as their prominent clinical presentation [27]. The family presented mild neurological symptoms along with neuropsychiatric symptoms, such as depression and schizophrenia, which could not be confirmed as primary or secondary manifestations of FD. Nevertheless, the authors lacked an alternative diagnosis and reported reduced α-galactosidase activity, together with increased levels of lyso-Gb3 in urine and plasma, and a pathogenic mutation in the GLA gene. The study does not report whether a whole genome analysis was carried out, which could rule out the contribution of alternative genes to this phenotype.

Additional psychiatric manifestations in patients with FD have been documented, albeit rarely, including acute psychotic symptoms, and personality and behavioral changes [24,28]. Another study focused on delineating a psychiatric and cognitive phenotype in FD in terms of psychiatric and cognitive functioning that was not only related to the difficulty these patients have coping with a chronic long-term disease [29].

The literature from some years ago suggested that there may be preservation of general intellectual functioning, memory, naming, perceptual functioning and global cognitive functioning in the absence of severe cerebrovascular events such as stroke or dementia [24]. In this same systematic literature review, they also pointed to evidence of impairment in executive functioning, information processing speed and attention. The more recent literature supports the principle of executive dysfunction in adult FD patients, with symptoms of attention-deficit/hyperactivity (ADHD), manifesting as difficulties with cognitive functioning, particularly in the realms of attention and concentration [30].

Cognitive impairment has been reported to be present in FD patients [24], and although a recent short-term follow-up study found no major changes in cognitive functioning, they studied the cohort of patients for only 1 year [31].

With regard to psychological impacts on pediatric patients, a study indicated that children with FD experience a poorer quality of life than their healthy counterparts [32]. Their results consistently identified adolescents with FD as being more heavily impacted than younger children, although not to the same degree as adults with FD.

A study in a Dutch cohort of 154 FD patients was not able to confirm the relationship between a history of stroke and depressive symptoms or between white matter lesions and depressive symptoms [32]. This further strengthens the hypothesis that brain abnormalities are not the main cause of depressive symptoms in patients with FD, and further studies are needed to elucidate the molecular and biochemical basis at a cellular level to understand the link between neuronal dysfunction and clinical manifestations.

Finally, focusing on sleep disturbances in FD, there are published data reporting a high prevalence of sleep-disordered breathing and abnormal periodic limb movements. A study showed that although the presence of abnormal periodic limb movements alone might have a minimal impact on sleep disturbance, they were associated with depression and analgesic requirements [33]. This supports the idea that exploring quality of sleep might also help FD patients and lessen the impact of FD with regard to psychological states.

## 4. MRI Abnormalities (Other Than Cerebrovascular Disease) in FD

While computed tomography (CT) application is limited to only acute cerebrovascular events, conventional MRI is the “gold standard” imaging technique to evaluate brain alterations in FD. The major conventional imaging findings in FD are (1) white matter hyperintensities (due to small vessel microangiopathy), which are the most common neuroradiological findings and present in up to 80% of patients; (2) stroke; (3) vertebrobasilar diameter abnormalities, which are a common, although inconstant, neuroradiological feature; and (4) the pulvinar sign, which was originally thought to be a common and pathognomonic sign of FD, but is no longer considered as such due to its low incidence and specificity [34,35].

Although it is the first-choice imaging technique, brain MRI fails in terms of diagnostic accuracy and thus has a poor sensitivity and specificity [34]. Brain lesions in FD are rather diffuse and do not have any specific anatomical localization [36]. Nevertheless, some new MRI techniques such as diffusion tensor imaging (DTI) have evidenced a good correlation with cognition (processing speed) and clinical disease severity [36]; they confirmed widespread areas of microstructural white matter disruption beyond the white matter hyperintensities seen on conventional MRI.

In the next sections, an analysis of the main findings with advanced neuroimaging techniques is discussed, with a focus on the possible link between them and the pathophysiology of the disease beyond its cerebrovascular involvement. There is also a brief comment on the neuroradiological findings in pediatric FD patients.

### 4.1. Reduced Intracranial Volume and Thalamic and Hippocampal Atrophy

The presence of cerebral atrophy (with losses in grey matter (GM) and white matter (WM)) in the absence of a severe cerebrovascular disease has been previously reported as a possible neuroradiological feature of FD, although there are some reported technical limitations in brain tissue volume studies [34,37]. Moreover, some focal differences in GM volume have been investigated, but no significant differences were found. Despite these previous findings, atrophy in specific brain regions, specifically the thalamus and the hippocampus, has been reported in FD patients compared to healthy controls and corrected for cerebrovascular events [34]. In addition, a global reduction in intracranial volume has been observed, suggesting the presence of abnormal neural development [38]. Clinical correlations with all these abnormalities have not been addressed, and additional studies are needed to investigate, for example, the role of the thalamus in pain perception, hippocampal atrophy in cognitive and memory complaints, or decreased global intracranial volume in abnormal brain development.

### 4.2. Motor Cortex and Cerebellar and Nigrostriatal Pathway Involvement

In a recent study using resting-state functional MRI (RS-fMRI), the presence of functional connectivity (FC) alterations in the motor circuits in patients with FD was evaluated [39]. FD patients with a history of stroke were excluded from the study, both male and female patients were included, and the patients were compared to healthy controls. There was significant FC involvement of the bilateral caudate and lenticular nuclei, as well as cerebellar involvement that encompassed portions of lobules 8 and 9 of both the vermis and cerebellar hemispheres. Functions of the basal ganglia and cerebellum have been related to the control of movement, but they also play a central role in processing cognitive and emotional information. These findings could shed some insights into the cerebral implications of FD, both in terms of symptoms related to PD and cognitive involvement.

Regarding basal ganglia involvement, a diffusion tensor imaging study showed the presence of microstructural damage affecting the thalamus [40].

In a study of three different pedigrees of FD patients who also exhibited signs and symptoms of akinetic-rigid PD, an 18F-DOPA-PET scan was performed [21]. There was evidence of reduced presynaptic dopaminergic enhancement in the nigrostriatal regions of these three patients, similar to what might be expected in idiopathic PD. These findings might support the hypothesis that the dopaminergic pathway is affected in FD patients. Moreover, another study revealed reduced nigral volume (suggesting neurodegeneration in this region) that correlated with the increased susceptibility of this region in FD patients [41].

### 4.3. Neurodegeneration, Neuronal Dysfunction and Hypometabolic Brain Regions

Brain MR spectroscopy (1H-MRS) has been used to investigate possible changes in the N-acetylaspartate/creatine (NAA/Cr) ratio (which might indicate neuronal degeneration and loss and is considered a marker of neuronal dysfunction) [42]. This study revealed diffuse reductions in the NAA/Cr ratio in different brain areas, affecting both cortical and subcortical structures. Nevertheless, these findings were inconsistent, as they were not reproducible in other studies [43].

In a recent review of quantitative susceptibility mapping investigating different neurodegenerative diseases (including one patient with FD), the authors found increased magnetic susceptibility in the putamen, caudate nuclei and substantia nigra in patients compared to controls [44]. Excess iron deposition in particular regions of the brain has been proposed as playing an important role in the pathology of neurodegenerative diseases, and whether this is related to neurodegeneration in FD has yet to be elucidated, since larger cohorts are needed and this is only an anecdotical, yet interesting, finding.

Regarding brain metabolism studied using positron emission tomography [45], hypometabolic areas were found only in regions with infarcts or hemorrhages on MRI scans, and there were no significant global glucose metabolic changes affecting the brains of FD patients [43].

### 4.4. MRI Changes in Children and Adolescents with FD

Children and adolescents with FD might present with brain MRI abnormalities in the form of white matter lesions (WML), deep grey matter lesions and infratentorial involvement [46]. In this study, they had a sample of 44 patients (20 boys and 24 girls, aged between 7 and 21 years old), and 90.9% were symptomatic of FD (neuropathic pain, cornea verticillata, abdominal pain or proteinuria). None of the patients showed microbleeds or any vascular abnormalities. A total of 7 out of 44 patients (15.9%) presented WML (5 girls and 2 boys), all patients presented the classic phenotype, and 3 patients (42.8%) had been receiving enzyme replacement treatment (ERT) for a mean period of 11 months. This frequency of abnormalities found in FD pediatric patients is higher than expected for the healthy pediatric population, which might have incidental and asymptomatic findings in brain MRI, with a reported frequency of 2.9–5.6%. It is also important to comment on these findings being present in three children who were receiving ERT. Studies regarding white matter hyperintensity progression while on ERT have been inconclusive; some studies reported progression despite treatment [47], and other studies ruled out a possible association between white matter hyperintensity (WMH) progression and ERT [48].

## 5. Neuropathology

Neuropathologic studies have shown that glycosphingolipid storage in FD is not exclusively present in the vasculature, but also in the neurons of brain regions known to be affected by different neurodegenerative conditions, such as the dorsal motor nucleus of the vagus, substantia nigra, and neocortex [49,50]. In a more recent report from Del Tredici et al., they found severe neuronal loss in the substantia nigra pars compacta and α-synuclein-immunopositive Lewy pathology. In this same study, they found Gb3 inclusions confined to somata, which frequently had a swollen or ballooned aspect, and were not seen in dendritic, axonal or intranuclear compartments, and the same Gb3 accumulations were also found in some astrocytic regions of the brain. This might suggest that CNS involvement in FD is not restricted to cerebrovascular events but involves alterations in different cell types, including both neurons and glial cells.

The hippocampus is a structure involved in psychiatric illnesses, e.g., hippocampal volume deficits have been found in both depression and schizophrenia, and this structure has been reported to be affected in FD. Postmortem studies have shown Gb3 accumulation in neurons in selective brain areas, including the hippocampus [27].

## 6. Animal Models

Mouse and rat models are available to test new treatments and elucidate mechanisms of disease pathogenesis [9], and in the next sections, a brief summary of the main findings is provided.

### 6.1. Mouse and Rat Models and the Recapitulation of FD Phenotype

The first published report of an FD mouse model (*B6;129-Gla^tm1Kul^/J*) was in 1997, being the animal model most used to date, and they reported that while Fabry mice appeared clinically normal at 10 weeks of age, microscopic and biochemical evidence of glycosphingolipid storage was evident [51]. Nevertheless, not even aging mice recapitulated the classic clinical signs observed in FD. In contrast, another study was indeed able to demonstrate alterations in sensorimotor function and hypoalgesia in Fabry mice, with a concomitant Gb3 accumulation in the peripheral nervous system, similar to that exhibited in patients [52]. Some years later, a rat model was generated using CRISPR/Cas9 technology to delete the rat *GLA* gene. It was demonstrated that Fabry rats recapitulated cardiorenal phenotypes and developed pain-like behavior similar to that observed in human patients [9].

In a study carried out with an FD mouse model vs. wild-type mice, affective and cognitive behavior was studied (with a focus on anxiety, depressive-like behavior and learning behavior) [53]. They did not find major differences between Fabry KO and WT mice and concluded that a major genetic influence on such symptoms in FD cannot be supported. It is worth noting that this study might also bring attention to the fact that the KO mice might not exhibit the same clinical phenotype as patients. In fact, as mentioned before, this model from Ohshima et al. recapitulates a less severe clinical phenotype, close to a late onset FD and exhibiting Gb3 accumulations only in the dorsal root ganglion neurons but not in the brain, as happens in humans [54,55].

In another study also carried out with an FD mouse model vs. wild-type mice, the pain response to temperature stimulus was studied [56]. They report that small type-C nociceptors from FD hemizygous mice exhibit mice exhibit a significant increase in the expression and function of the TRPV1 (thermoTRP) channel, implicated in a painful heat sensation. This might partly explain the molecular mechanisms underlying one of the main symptoms experienced by FD patients: The neuropathic pain that appears in the early stage of the disease as a result of peripheral small fiber damage.

### 6.2. Alterations in the Autophagy–Lysosome Pathway (ALP)

The ALP, which is a common hallmark of LSD that may be partly implicated in the onset and progression of nervous system pathophysiology, was studied in the AGA deficiency mouse model and compared to wild-type mice [57]. Different ALP markers in brain specimens from the mouse model using immunofluorescence analysis and electron microscopy were evaluated. The ALP is considered to be an important signaling pathway that maintains the intracellular energy balance and affects cell survival [58], and although alterations in the ALP had been previously reported in muscle, kidney and fibroblasts from FD patients, this was the first report of such alterations affecting the CNS in a mouse model. They demonstrated the immunoreactivity of different ALP markers localized to perinuclear and neuritic regions of neurons and detected axonal spheroids in the pons, possibly indicating axonal degeneration [57].

### 6.3. PD and Brain Protein Aggregation

Regarding PD, a recent publication aimed to study the accumulation of aggregated α-synuclein, considered to be the pathological hallmark of PD and related synucleinopathies [59], in neurons from different symptomatic LSD mouse models, including FD [60]. Lysosomal dysfunction in particularly susceptible neurons might result in aberrant processing and the misfolding of proteins such as α-synuclein. The authors evaluated the presence of aggregated protein pathology and inflammation in the CNS and found evidence of proteinopathy in all LSDs with the exceptions of FD and Gaucher disease, for which no signs of neuroinflammation could be found. Tau protein and alpha-synuclein were detected in the hippocampus and cerebellum of the FD mouse model but were absent in the cortex and brainstem.

### 6.4. Gene Expression Related to Neuronal and Synaptic Dysfunction

Another study explored neuronal gene expression changes in the prefrontal cortex in tissue samples from adult hemizygous AGA knock out male mice [61]. They performed mRNA microarray expression profiling, followed by qPCR validation and in-depth bioinformatics analyses of protein–protein interactions and pathways. In addition to a variety of up- or downregulated genes related to inflammatory and immune responses and the regulation of cytokine production, they specifically screened for genes related to ion channels, receptors, synapses and signaling proteins. They found that some genes were upregulated (e.g., the nicotinic acetylcholine (ACh) receptor α-subunit 4, among others), and some genes were downregulated (e.g., two potassium channels and synaptic proteins such as synaptojanin-2 and other proteins from the active zones). The study concluded that such findings suggested possible functional changes in the brain as a consequence of AGA depletion that may affect synaptic signaling and information processing in cortical circuits. This study is of particular interest considering that the prefrontal cortical region was the region of interest and that this brain area is associated with cognitive deficits as well as different executive functions.

### 6.5. New Treatment Options with Impact on the CNS

The only specific treatment that it is available up to date for FD, which can cross the BBB is the pharmacological chaperone Migalastat; however, its benefits on neurobehavioral involvement on FD have not been systematically studied and reported up to date.

Regarding new possible treatment options that might have an impact on the CNS in FD patients, a newly reported gene-based therapy was able to cross the blood–brain barrier (BBB) in a preclinical study on an FD KO mouse model. The therapy consists of the systemic administration of an adeno-associated viral vector 9 (AAV-9) that mediates widespread *GLA* expression and function in the CNS and in multiple tissues, and prevents glycosphingolipid accumulation, when it is administered to both presymptomatic and symptomatic animals. This finding are very promising, but the efficacy of this gene transfer approach still has to be demonstrated in clinical trials. In case this approach will be successful, it can be a concrete option to reach the CNS in FD without using invasive methods, which is an interesting aim, since the currently available ERT drugs cannot cross the BBB and the chaperone is not indicated for all FD patients [54].

## 7. Induced Pluripotent Stem Cell Models in FD

The recently introduced technology of induced pluripotent stem cells (iPSCs) derived from a patient’s biopsy have provided new opportunities to explore the cell biology and pathophysiology of human diseases, as well as the assessment of new therapeutic strategies. Through reprogramming and differentiation of iPSC, it is possible to obtain patient models of functional cells and tissues to gain insight into disease etiology and the possible approaches to restore the impaired mechanism.

Patient-derived iPSC models have been generated for a number of LSD, including Gaucher disease, Pompe disease, metachromatic leukodystrophy, the neuronal ceroid lipofuscinoses, Niemann–Pick types A and C1, several of the mucopolysaccharidoses and also FD [62].

An in vitro model which recapitulated the clinical features of FD cardiomyocytes was generated by reprogramming iPSC [63]. This model shows accumulating Gb3 and displays an increased excitability, with altered electrophysiology and calcium handling. Using proteomics technology to analyze these patient’s derived cardiomyocytes revealed the accumulation of the lysosomal protein LIMP-2 and the secretion of cathepsin F and HSPA2/HSP70-2 in FD, which can be reverted by the genetic editing of the *GLA* mutation [64].

Nevertheless, the generation of neural stem cells (NSCs) and neurons, which efficiently recapitulates the features of FD in the CNS, has not been completely achieved at the moment. In spite of the success of this technology in other LDs [65], a proper model for FD neurons was still not obtained. Miyajima et al. were able to successfully derive NSCs from an FD patient’s iPSCs; however, cellular damage and morphological changes similar to the ones found in the patient’s brain were not recapitulated in these cells, where no Gb3 accumulation was detected [66].

Further efforts to mimic in culture the in vivo environmental conditions are required, to obtain neurons that effectively reproduce disease-specific features. FD NSCs will provide a valuable resource to understand the possible involvement of AGA in CNS functional regulation and the link between *GLA* and neurodegenerative conditions, such as Parkinson’s disease.

## 8. Energy Metabolism and Neurotransmission in FD

### 8.1. Energy Metabolism and Neurotransmission in the CNS, Lysosomal Disorders and Glycosphingolipodoses

Glucose is the main energy substrate for the adult brain, while there are other alternative substrates that are required for the developing brain, such as ketone bodies, glycerol, lactate, amino acids and fatty acids. All these substrates are utilized for energy production, the biosynthesis of lipids and proteins, and synaptic transmission (which is the most demanding energy process in the adult brain) [67,68]. Energy production and appropriate functioning of the tricarboxylic acid (TCA) cycle are of utmost importance in regard to the synthesis of neurotransmitters and other synaptic and neuronal compounds. The TCA cycle in the CNS is specifically supported by the replenishment of its intermediates generated in astrocytes, giving special relevance to the communication and interaction between neurons and astrocytes [67], the two main cellular components in the CNS devoted to neurotransmitter synthesis (especially GABA and glutamate).

Lysosomes are organelles not only involved in the breakdown of complex molecules (i.e., lipids, carbohydrates, proteins and nucleic acids) but also involved in plasma membrane repair, nutrient and oxidative stress sensing for the cell, the homeostasis of calcium and copper, and processes of endocytosis, phagocytosis and autophagy [69,70]. Lysosomal dysfunction can alter lipid composition at contact sites between mitochondria and the endoplasmic reticulum, impairing the reciprocal communication between these organelles. Focusing on glycosphingolipidoses (GSLs) (FD has a role in GSL degradation), GSLs have an active role in controlling the architecture of lipid rafts and, therefore, in regulating the types of intermembrane interactions and the membrane composition with different cell adhesion molecules, ligands, receptors, and other molecules present in the cell membrane [71].

Moreover, lysosomal dysfunction can alter neurotransmitter and neurotransmission homeostasis, since there is a putative role for lysosomes in the turnover and modulation of synaptic protein composition [67]. For all these reasons, there is increasing evidence of the implications of lysosomal dysfunction interfering in energy production and in maintaining synaptic homeostasis.

Given all this evidence, there are two main consequences of lysosomal dysfunction at two different levels: (1) in the organelle itself, there is an altered interaction between the lysosome and the other organelles; and (2) in the cell, there is an altered membrane composition with potential implications for cellular signaling and interactions, energetic metabolism related to mitochondrial functioning, etc.

Nevertheless, little is known about the implication for FD in all these mentioned roles, although they have been widely studied in the more common neurodegenerative lysosomal disorders, such as mucopolysaccharidosis (MPS IIIA and IIIB), Batten disease and other neuronal ceroid lipofuscinoses, such as Niemann–Pick type C, gangliosidosis and Krabbe disease. In fact, with a focus on synaptic dysfunction, different implications of LSDs have been identified and include (1) impairments in endocytosis and exocytosis (affecting vesicle trafficking and the membrane composition of the different SNARE proteins); (2) the development of axonopathy; (3) changes in synaptic proteins (i.e., VAMP2, synaptophysin and syntaxin, among others); (4) alterations in the generation and recycling of synaptic vesicles; (5) defects in synaptic spines; and (6) changes in postsynaptic density [7].

In the case of FD, individual studies have reported different alterations that involve different neurotransmitter systems, but no consistent studies have been performed up to date.

### 8.2. Mitochondrial Involvement, Impaired Autophagic Function and Altered Neurotransmitter Release in FD

As previously mentioned, lysosomes interact with mitochondria, the central organelle devoted to ATP and energy production. In a publication from Kaneski et al., they developed an in vitro model system to study neuronal dysfunction in FD by creating a stable knockdown of AGA in a human cholinergic neuronal cell line [72]. They proved that this cellular model reproduces the pathophysiology of FD well in terms of reduced AGA activity and the accumulation of the stored substrate Gb3. Given this evidence, they tested both basal release and the potassium-stimulated release of ACh in the silenced cells and found that neurotransmitter release was significantly altered in these cells. There are at least two main hypotheses to explain such a reduction in Ach release: (1) the previously mentioned role of lysosomes in controlling lipid raft and membrane protein composition and distribution, which could influence synaptic vesicle formation and fusion with the plasma membrane or (2) decreased ACh synthesis in the mitochondria. Regarding this second hypothesis, ACh is synthesized in neurons from choline and acetyl-CoA (the latter is produced in mitochondria as a product of the citric acid cycle and transported into the cytoplasm for ACh synthesis), and ACh is then packed into the synaptic vesicles. Altered mitochondrial function might lead to a decrease in ACh synthesis due to altered acetyl-CoA availability and a reduction in the necessary ATP supply to assist neurotransmission (energy supply is one of the main pillars of neurotransmission) [68]. In a previous study, altered mitochondrial energy production was demonstrated in fibroblasts from FD patients [73]. They demonstrated that respiratory chain enzyme activities I, IV and V were significantly (*p* < 0.01) lower in FD cells, with a subsequent drop in the cellular levels of energy-rich phosphates (ATP and derivatives).

Ivanova et al. explored the links between lysosomal abnormalities in Gaucher disease and FD, the ALP, and the regulation of cellular energy homeostasis in peripheral blood mononuclear cells derived from patients. They demonstrated that lysosomal abnormalities, independent of the type of accumulated substrate, suppressed not only autophagy but also mitochondrial function and mTOR signaling pathways [74]. ERT partially restored these functions and improved mitochondrial markers.

### 8.3. Different Neurotransmitter Systems Altered in FD, and Their Possible Role in Clinical Manifestations

Very little has been published regarding any specific neurotransmitter system altered in FD and the resulting clinical implications. In the next sections, there is a brief summary of the little evidence available in the literature and a few hypotheses regarding its clinical implications.

#### 8.3.1. Acetylcholine (ACh)

The implication of disrupted ACh neurotransmission has been previously discussed in the study by Kaneski et al., although from a peripheral nervous system perspective. They postulated that since cholinergic neurons are associated with eccrine sweat glands, blood vessels, hair follicles, and cutaneous sensory nerve endings (all of which are affected in FD), this might be the link between impaired ACh release and neuropathic pain and hypohidrosis, which are hallmarks of the disease [72].

#### 8.3.2. Dopamine (DA)

Presynaptic dopaminergic disruption has been reported in a neuroimaging study with 18F-DOPA PET scans performed in patients with FD and PD [21]. They described dopaminergic presynaptic deficits both in the nigrostriatal and striatal regions in three different patients, which were similar to the deficits expected to be found in primary PD.

#### 8.3.3. Dopamine and Acetylcholine

An imbalance between cholinergic activity and dopaminergic activity in the basal ganglia (especially the striatum) has been widely explored in the context of movement disorders and has been identified as causing a variety of neurological disorders, such as PD [60]. Moreover, the loss of cholinergic transmission in Alzheimer’s disease and the loss of midbrain dopaminergic neurons in PD has been studied in specific neuronal populations in these two neurodegenerative disorders [75]. In the case of FD, isolated Gb3 accumulations were identified in the nerve cells of two cholinergic nuclei in the basal forebrain (the basal nucleus of Meynert and bed nucleus of the diagonal band), and severe atrophy in the substantia nigra pars compacta was described [23].

#### 8.3.4. Serotonin (5-HT)

The serotonergic system extends widely throughout the CNS, it is known to modulate a broad spectrum of functions (including mood, cognition, anxiety, learning, memory, reward processing, and sleep), and deficits in the serotonergic system can result in various disease conditions, particularly depression, schizophrenia, mood disorders, and autism [76]. Modulation of the serotoninergic system is complex, and there are possibly different aspects related to disease manifestations; in the case of FD, large Gb3 accumulations in serotonergic nuclei in the lower and upper raphe groups have been described [23]. A previously mentioned study reported hippocampal atrophy [27], which is a structure known to be related to stress, memory consolidation and major depressive disorder, and receives input from the substantia nigra par compacta, which in turn is modulated by serotonin [77,78]. Whether Gb3 accumulation and hippocampal atrophy are directly linked to psychiatric manifestations needs to be further explored.

## 9. Conclusions

The literature on FD and CNS involvement is scarce when focusing on alterations beyond cerebrovascular disease and is nearly absent in regard to synaptic dysfunction. More studies are needed to identify some of the debilitating and long-term clinical implications of FD as experienced by patients, such as depression, PD, anxiety or low quality of life. This is important to highlight the necessity of a multidisciplinary team caring for these patients that is not only focused on the main organs that are determinant for their life expectancy (i.e., kidney and heart) but also focused on their psychopathological state and early indicators of Parkinsonism, depression or dementia. There is an urgent need for basic research on FD using cellular and animal models that could help clarify the pathophysiology of synaptic involvement in this lysosomal disorder. Even if a clear involvement of the substantia nigra pars compacta and other CNS-related manifestations still have to be clearly demonstrated in FD, it is important to consider that, as we recapitulate in this article, at least milder CNS functional impairment could be contributing to affect the clinical course of patients today, who present with higher survival rates compared to those first-diagnosed subjects, thanks to the currently available treatments. In this sense, we believe that more studies on the neuro-behavioral effects of FD are required to evaluate the opportunity of using specific treatments that can cross the BBB to ameliorate CNS-related manifestations.

## Figures and Tables

**Figure 1 ijms-24-05246-f001:**
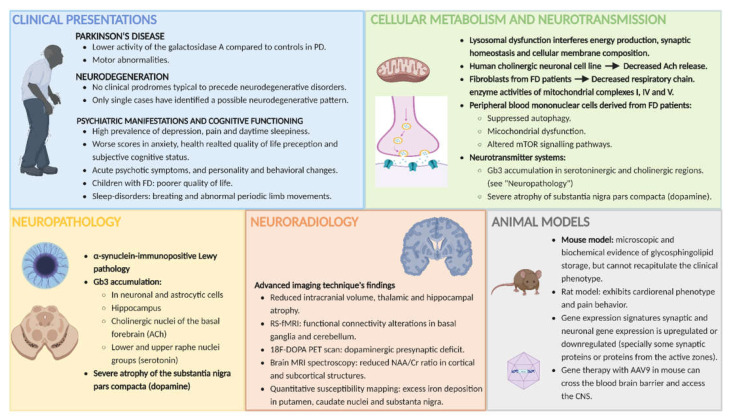
Graphical summary of the central nervous system implications in Fabry disease. This figure summarizes the main findings that will be highlighted and discussed in this review. It focuses on the central nervous system implications in Fabry disease that go beyond the classical involvement of the peripheral nervous system and cerebrovascular disease. The picture addresses the different points of view on FD and its clinical implications, as well as its radiological and neuropathological findings, and cellular, metabolic and synaptic implications (follow main text for details in the next sections). Abbreviations: AAV9 = adeno-associated Virus 9; ACh = acetylcholine; FD = Fabry disease; Gb3 = globotriaosylceramide; mTOR = mechanistic target of rapamycin; NAA/Cr ratio = N-acethyl aspartate/creatine ratio; PD = Parkinson’s disease; PET = positron emission tomography; RS-fMRI = resting-state functional magnetic resonance imaging.

## Data Availability

The literature that was analyzed for this review is available in Pubmed.gov.

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
