# Peer review of "Fabry Disease and Central Nervous System Involvement: From Big to Small, from Brain to Synapse"

_ijms, 2023, doi:10.3390/ijms24065246_

Round 1

Reviewer 1 Report

Dear editor:

Thank you for inviting me to evaluate the review article titled “Synaptic and Neuronal Dysfunction in Fabry Disease: What We Know”. In current manuscript authors summarized the synaptic and neuronal dysfunction in FD from 6 sections which include clinical manifestation, MRI abnormalities, neuropathology, animal models, energy metabolism, and neurotransmission. In general, this manuscript could be considered published on IJMS. However, there are some comments as below should be modified. Authors should prepare a major revision for publish in second review.

1.   FD is a lysosomal storage disorder (LSD) with X-linked 25 inheritance secondary to mutations in the GLA gene. What’s the differences between men and women? Authors should describe them in their article.

2.   Authors mentioned “FD are considered to be small fiber neuropathy associated with pain”. Pain is one of the earliest clinical symptoms in Fabry disease (FD) reported by children and young adults. Neuropathic pain, defined as that arising from a lesion or disease of the spinal cord and/or brain is a key feature of the disease. Please carefully review the manuscript and add more details of the pain in FD.

3.   MRI findings associated with FD include white matter hyperintensities (WMH) and cerebral microbleeds (CMBs). What’s the limitation of MRI in FD?

4.   Authors gave a summary of models available and used in the field of FD in animal models in general. How do authors think about the vitro cell models? To date, most tissue engineering strategies rely on established cell lines (often transformed cell lines) or primary cells derived from patients. Like Human-induced pluripotent stem cells (iPSCs) models in FD.

Base on above mentioned, authors should resubmit a major revision.

Author Response

Dear editor:
Thank you for inviting me to evaluate the review article titled “Synaptic and Neuronal Dysfunction in Fabry Disease: What We Know”. In current manuscript authors summarized the synaptic and neuronal dysfunction in FD from 6 sections which include clinical manifestation, MRI abnormalities, neuropathology, animal models, energy metabolism, and neurotransmission. In general, this manuscript could be considered published on IJMS. However, there are some comments as below should be modified. Authors should prepare a major revision for publish in second review.

1.   FD is a lysosomal storage disorder (LSD) with X-linked inheritance secondary to mutations in the GLA gene. What’s the differences between men and women? Authors should describe them in their article.
We thank the reviewer for this comment. We addressed this issue and added some more information on this topic at the introduction of our manuscript. 

2.   Authors mentioned “FD are considered to be small fiber neuropathy associated with pain”. Pain is one of the earliest clinical symptoms in Fabry disease (FD) reported by children and young adults. Neuropathic pain, defined as that arising from a lesion or disease of the spinal cord and/or brain is a key feature of the disease. Please carefully review the manuscript and add more details of the pain in FD.
Thank you for this very important remark. It is true that pain in one of the main hallmarks of the disease. We have not initially included the pain in our manuscript, as it was intended to be focused only in the central nervous system. Since neuropathic pain is mainly due to disturbances in the peripheral nerve and the dorsal root ganglion, we have considered this to be beyond the main scope of our review. Nevertheless, and as the reviewer suggests adding some of this topic in our review, we added a new paragraph (“3.1. The pain in Fabry disease”) and tried to focus on what was found about pain and its impact on the brain. 

3.   MRI findings associated with FD include white matter hyperintensities (WMH) and cerebral microbleeds (CMBs). What’s the limitation of MRI in FD?
Thank you for this remark. We have added some more comments on the current limitations of MRI (in terms of sensitivity and accuracy in diagnostic) and added a new study aiming to correlate DTI imaging with cognition disturbances in FD. 

4.   Authors gave a summary of models available and used in the field of FD in animal models in general. How do authors think about the vitro cell models? To date, most tissue engineering strategies rely on established cell lines (often transformed cell lines) or primary cells derived from patients. Like Human-induced pluripotent stem cells (iPSCs) models in FD.
We believe that this comment is very pertinent and we thank the reviewer for helping us improving the quality of the manuscript. iPSCs models are gaining importance in studying LSDs pathophysiology and possible treatment strategies.
In the revised manuscript we included a session entitled “Induced pluripotent stem cells models in FD”, where we summarize the features of iPSCs derived cellular models available for FD.

Reviewer 2 Report

For this manuscript, the authors summarize the literature about Fabry disease, focusing on its effect on the central nervous system. The manuscript is well organized and the authors covered recent literature on Fabry disease. Nevertheless, I have some minor comments.

  1. The authors mainly focus on the central nervous system, including synaptic and neuronal dysfunctions, but not limit to them. Thus the title seems misleading.

  2. In the manuscript, there are many long sentences that are hard to follow, such as this one in the abstract “The literature on FD and central nervous system involvement is scarce when focusing on alterations beyond cerebrovascular disease and nearly absent in regard to synaptic dysfunction, although reports have provided evidence for CNS clinical implications of FD, including Parkinson’s disease, neuropsychiatric disorders and executive dysfunction.” The authors should spend more time trying to make the sentence simple and clear.

  3. The section “The Framework of LSDs” may be redundant.

  4. Line 91, after “cerebrovascular disease”, we should have a reference.

  5. Line 123, “or for example”, whether “or” is redundant?

  6. What is the criteria of reference searching? Some references are missing from the list, such as this one: PMID: 17279083. 

Author Response

Dear Editor,

We thank you and the two reviewers for the very constructive comments.

Below we have addressed the comments on a point-by-point basis. Each point starts with the quote of the original comment followed by our response (blue text). All co-authors have agreed to these changes prior to submission.

We hope that we have addressed all comments satisfactorily and that you will consider the revised manuscript worthy of publication.

Yours sincerely,

Saida Ortolano, correspondig author

  1. The authors mainly focus on the central nervous system, including synaptic and neuronal dysfunctions, but not limit to them. Thus the title seems misleading.

The reviewer is right and we really appreciate this opinion. We changed the title to adjust the message of our manuscript.

Old title: “Synaptic and neuronal dysfunction in Fabry Disease: what we know”.

New title: “Fabry disease and central nervous system involvement: from big to small, from brain to synapse”.

  1. In the manuscript, there are many long sentences that are hard to follow, such as this one in the abstract “The literature on FD and central nervous system involvement is scarce when focusing on alterations beyond cerebrovascular disease and nearly absent in regard to synaptic dysfunction, although reports have provided evidence for CNS clinical implications of FD, including Parkinson’s disease, neuropsychiatric disorders and executive dysfunction.” The authors should spend more time trying to make the sentence simple and clear.

We sent our initial manuscript for English correction in terms of style and content. We did so through the services of a prestigious editorial and journal. Nevertheless, it is true, as the reviewer says, that some sentences are long and might be difficult to follow. For this reason, we went through the text again and stated some sentences more clearly by putting them shorter.

  1. The section “The Framework of LSDs” may be redundant.

We thank the reviewer for this remark. We reframed this section, made it shorter and more directly focused on FD and not the other LSD.

  1. Line 91, after “cerebrovascular disease”, we should have a reference.

Thank you for your proposal. We addressed that change, and added a reference there as suggested.

  1. Line 123, “or for example”, whether “or” is redundant?

Thank you for your proposal. You were right, and we deleted the redundant word “or”.

  1. What is the criteria of reference searching? Some references are missing from the list, such as this one: PMID:

It is true that this is an important reference in a high impact factor journal, but this is from 2007 and we initially only included publications from 2009 and beyond. But, given that we also added some more references based on the search of the reference lists in every relevant paper, we can add this reference as well (and we did so in fact, when we addressed point number 4).

We would like to remark, that since the manuscript was not planned as a systematic review (meaning that every reference was not categorized and rated based on standardized methodologies such as GRADE) we have not added a “Methods and Results” section. However, we are open to adopt such an interesting proposal, as we consider that doing this would add value to our manuscript. We aim here to clarify how the literature review was conducted, and we present this description in the supplementary information section.

Round 2

Reviewer 1 Report

None.